# A machine learning approach
# that beats Rubik's cubes

**Alexander Chervov**[*]
Institut Curie, Universite PSL
Paris, F-75005, France

**Kirill Khoruzhii**[*]
Technical University of Munich
Garching, 85748, Germany

**Nikita Bukhal**
Novosibirsk State University
Novosibirsk, 630090, Russia

**Jalal Naghiyev**
Technical University of Munich
Garching, 85748, Germany

**Vladislav Zamkovoy**
Independent Researcher
Moscow, 119454, Russia

**Ivan Koltsov**
RTU MIREA
Moscow, 119454, Russia

**Lyudmila Cheldieva**
RTU MIREA
Moscow, 119454, Russia

**Arsenii Sychev**
RTU MIREA
Moscow, 119454, Russia

**Arsenii Lenin**
RTU MIREA
Moscow, 119454, Russia

**Mark Obozov**
Innopolis University
Innopolis, 420500, Russia

**Egor Urvanov**
RTU MIREA
Moscow, 119454, Russia

**Alexey M. Romanov**[†]
RTU MIREA
Moscow, 119454, Russia
`romanov@mirea.ru`

## Abstract

The paper proposes a novel machine learning-based approach to the pathfinding problem on extremely large graphs. This method leverages diffusion distance estimation via a neural network and uses beam search for pathfinding. We demonstrate its efficiency by finding solutions for 4x4x4 and 5x5x5 Rubik's cubes with unprecedentedly short solution lengths, outperforming all available solvers and introducing the first machine learning solver beyond the 3x3x3 case. In particular, it surpasses every single case of the combined best results in the Kaggle Santa 2023 challenge, which involved over 1,000 teams. For the 3x3x3 Rubik's cube, our approach achieves an optimality rate exceeding 98%, matching the performance of task-specific solvers and significantly outperforming prior solutions such as DeepCubeA (60.3%) and EfficientCube (69.6%). Our solution in its current implementation is approximately 25.6 times faster in solving 3x3x3 Rubik's cubes while requiring up to 8.5 times less model training time than the most efficient state-of-the-art competitor. Finally, it is demonstrated that even a single agent trained using a relatively small number of examples can robustly solve a broad range of puzzles represented by Cayley graphs of size up to $10^{145}$, confirming the generality of the proposed method.

---

[*]Alexander Chervov and Kirill Khoruzhii contributed equally to this work.
[†]Corresponding author

39th Conference on Neural Information Processing Systems (NeurIPS 2025).

# 1 Introduction

Rubik's cube is one of the most famous puzzles, which is believed to be played by more than a billion people in the world [1]. According to [2], it was included in the 100 most influential inventions of the 20th century. Even decades after its first introduction, it is still used as a benchmark and model task in various fields: artificial intelligence [3], robotics [4], graphs algorithms [5], [6], cryptography [7], image encryption [8], statistical physics [9],[10], group theory [11],[12], for human cognitive abilities [13].

From a broader perspective, solving the Rubik's Cube is a specific instance of a planning problem. One must plan actions to transition between the initial and solved states. The mathematical framework for such problems is pathfinding on graphs (state transition graphs): all possible states are represented as nodes, and edges correspond to transitions between states based on actions (moves). The planning task thus reduces to finding a path from a given initial node to one or more desired nodes. A specific class of graphs represents the Rubik's Cube and similar puzzles—Cayley-type graphs of the puzzle's symmetry group. These are highly symmetric state transition graphs where the symmetry group can transform any node into another. Cayley graphs are of fundamental importance in modern mathematics [14], [15] and have numerous applications: in bioinformatics for estimating evolutionary distances [16, 17, 18, 19]; in processor interconnection networks [20, 21, 22]; in coding theory for the construction of expander graphs and related codes [23]; in cryptography for constructing specific hash functions [24, 7]; in machine learning (ML) [18]; and in quantum computing [25, 26, 27, 28, 29].

Finding the shortest paths on generic finite Cayley graphs is an NP-hard problem [30], as it is for many particular groups: the Rubik's Cube group [31] and some others [32, 18]. Brute force breadth-first search, Dijkstra's, and related methods can find the shortest paths on graphs with billions of nodes, the bidirectional trick squares feasible sizes, but these methods require extremely large computational resources and are not practical for much larger sizes, which are of our interest. Moreover, no effective tools are currently available to find any (not just the shortest) paths on Cayley graphs of large finite groups. For example, modern computer algebra systems like GAP [33] fail on any sufficiently large group, such as the 4x4x4 Rubik's Cube.

This research aims to overcome the abovementioned limits in solving large Rubik's cubes and similar large-scale pathfinding problems with a high level of optimality using machine learning. The main contributions to the state of the art are the following:

1. We propose a novel multi-agent, machine learning-based approach to find paths on Cayley graphs of finite groups. It is the first machine learning approach capable of handling groups as large as $10^{74}$. It achieves over 98% optimality on the DeepCubeA dataset of 3x3x3 cubes, reaching the level of task-oriented solvers based on pattern databases. It produces better results (shorter solution paths) than any known competitor for 4x4x4 and 5x5x5 Rubik's Cubes, including the aggregated best results from the 2023 Kaggle Santa Challenge, representing the current state of the art.

2. We demonstrate that increasing the size of the set used to train multilayer perceptrons with residual blocks has a limited impact on the pathfinder's performance. At the same time, increasing the beam width and number of agents robustly improves the average solution length and optimality. This surprising finding helped choose the size of the train data for each agent and achieve best-in-class performance without wasting computational resources on additional training.

3. The training time and computational resources required for our approach are significantly smaller than those for state-of-the-art approaches. Our solution, tested on the same hardware and beam width providing solution length similar to the EfficientCube (the previous leading ML solution) solving the task approximately 25.6 faster and requiring up to 8.5 times less model training time than the competitor.

4. We demonstrate that even a single agent trained using a relatively small number of examples can robustly solve a broad range of puzzles represented by Cayley graphs with sizes up to $10^{145}$, confirming the generality of the proposed method.

In recent years, machine learning has been emerging as "a tool in theoretical science" [34], leading to several noteworthy applications to mathematical problems [35, 36, 37, 38, 39, 40, 41, 42]. This research is part of the larger project, which aims to create an open-source machine learning Python

framework for analyzing Cayley graphs and contribute to the fascinating, emerging area of machine learning applications in theoretical sciences.

## 2 Proposed Machine Learning Approach

This paper presents a unified approach for finding paths on a large class of graphs, focusing on demonstrating its efficiency for Rubik's cube graphs. It does not rely on any prior knowledge or human expertise about the graphs. The approach has two main components: a neural network model and a graph search algorithm — similar to previous works such as AlphaGo/AlphaZero [43],[44], DeepCube [45],[3], and EfficientCube [46], among others. The model is trained to guide what moves should be done to get closer to the destination node ("solved state" for puzzles). The graph search algorithm starts from a given node and moves to nodes closer to the destination, based on the neural network's predictions, until the destination node is found.

The basic assumption on a graph is that there is a vector associated with each node (feature vector). These vectors serve as an input for the neural network. The precise quantification of requirements for feature vectors that would ensure the successful operation of the proposed method is challenging. We aim to demonstrate its efficiency in the context of Rubik's group cases. On one extreme, even random vectors suffice if the training data covers all nodes — an idea employed in well-known approaches such as DeepWalk [47] and Node2vec [48]. However, our focus is different: only a small subset of nodes will be covered by the training data (random walks). The key point is the ability of the neural network to generalize from that small subset to the entire graph — something that is impossible with random features. Worse, the feature vectors are related to the distance between nodes on a graph — more training data is required, and more advanced parameters and resources should be used at all steps of the proposed method. The role of the neural network is to transform the initial feature vectors into a latent representation, where nodes that are closer on the graph are also closer in the latent space. For puzzles or permutation groups the feature vector is just the vector describing the permutation $p$ of $l$-symbols, i.e. vector of numbers $(p(0), p(1), ..., p(l-2), p(l-1))$. Additionally, we assume that a specific node on the graph, such as the 'solved state' for puzzles, is selected. The task is to find a path from any given node to this selected node. Since the graph sizes may exceed $10^{40}$, standard pathfinding methods are not applicable.

The key steps of the proposed method are illustrated in the figure 1a and described below:

**Creating the training set via random walks. (Diffusion distance.)** Generate $N$ random walk trajectories starting from a selected node. (The generation of a random walk is a simple process: select a random neighbor of the current node and repeat this process iteratively for multiple steps.) Each random walk trajectory consists of up to $K_{\max}$ steps, where $N$ and $K_{\max}$ are integer parameters of the method. For some nodes encountered during the random walks, we store a set of pairs $(v, k)$, where $v$ represents the vector corresponding to the node and $k$ is the number of steps required to reach it via the random walk. This set will serve as the training data. For the Rubik's Cube, random walks correspond to random scrambling: starting from the "solved state," we perform a series of random scrambles and record the resulting positions and the number of scrambles performed. Conceptually, in the limit as $N \to \infty$, the average value of $k$ measures the "diffusion distance" — roughly speaking, the length of the random path or an estimate of how quickly diffusion reaches a given node. In contrast to the DAVI approach used in [3], random walk generation is very computationally cheap, making it possible to generate them directly during the training procedure.

**Training the neural network.** The generated set of pairs $(v, k)$ serves as the training set for the neural network. Specifically, $v$ serves as the 'feature vector' (the input for the neural network), and $k$ represents the 'target' (the output the network needs to predict). Thus, the neural network's predictions for a given node $v$ estimate the diffusion distance from $v$ to the selected destination node (solved state of the puzzle). We utilize a multilayer perceptron (MLP) architecture with several residual blocks and batch normalization, as shown in Figure 1b, which will be further called ResMLP. It is a general form of the MLPs used in [3, 46]. All the models are trained in advance before the solving phase.

**Graph search guided by a neural network. Beam search.** This step finds a path from a given node to the destination node. The neural network provides heuristics on where to make the next steps, while the graph pathfinding technique compensates for any possible incorrectness in the neural network predictions. The beam search pathfinding method is quite simple but has proven to be the

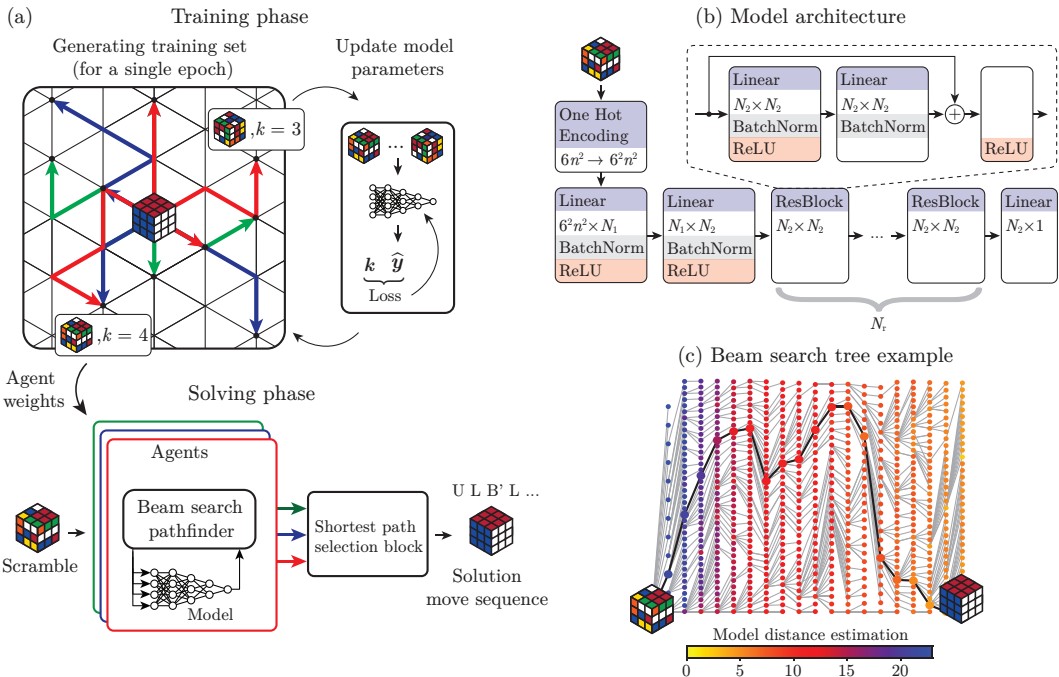

Figure 1: Proposed ML solution for Rubik's cube solving: (a) proposed multi-agent solver's process flow; (b) ResMLP neural network architecture; (c) an example of beam search pathfinding on 3x3x3 cube's graph using $W = 40$.

most effective for us and works as follows. Fix a positive integer $W$ — a parameter known as the "beam width" (or "beam size"). Starting from a given node, we take all its neighboring nodes and compute the neural network predictions for all of them. We then select the $W$ nodes closest to the destination according to the neural network (i.e., the predictions have smaller values). We take these selected $W$ nodes' neighbors, drop duplicates, and again compute the neural network predictions, choosing the top $W$ nodes with the best (i.e., minimal) predictions. The search iterations are repeated until the destination node is found (or the limit of steps is exceeded). The whole process is illustrated in Figure 1.

**Multi-agency.** The method described in the steps above relies on random walks for train set creation, and thus, due to that randomness, each new launch will create a new train set, and thus, each new neural network approximates the distance differently. This diversity is large enough to yield a new solution path for each launch typically. And hence, typically, several repetitions allow for the discovery of a shorter path than a single run. We call each trained neural network an agent. To solve any given state - we solve it with all the agents and then choose the best result (the shortest solution path among all the agents) – illustrated in Figure 1a.).

## 3 Optimality vs the Proposed Approach Parameters

The proposed solver has the following main parameters $A$ – the number of agents, $W$ – beam width used by each agent during pathfinding and ResMLP model general parameters: $N_1$ – the size of the first layer, $N_2$ – the size of the second layer and residual blocks' layers, $N_r$ – the number of residual blocks and $T$ – the trainset size. For easier comparison, $N_1$, $N_2$, and $N_r$ are also summarized by model size $P$ – the total number of ResMLP parameters (weights and biases). In this section, we analyze the influence of these parameters on the solver's average solution length and optimality.

**Train set size.** First, in the example of 3x3x3 and 4x4x4 Rubik's cubes, we analyzed how the model and trainset sizes, as well as model depth, influence the average solution length using a single agent with fixed beam width ($W = 2^{18}$). The experiment details are provided in Appendix B.6, while the results are presented in Figure 2a. It is seen from Figure 2a that from a certain point, the raise of

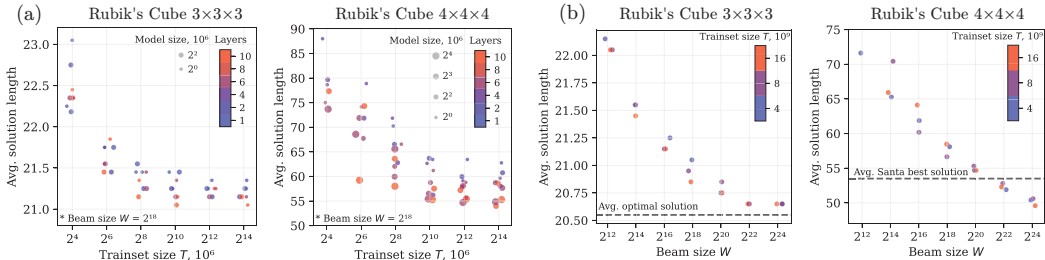

Figure 2: Influence of model parameters on solution length for 3x3x3 and 4x4x4 cubes (jitter plot): (a) influence of the model size, trainset sizes, and model depth on average solution length; (b) influence of the beam width on average solution length.

$T$ does not lead to any significant reduction of average solution length, especially considering the fact that the trainset size is demonstrated in logarithmic scale. Even more surprising, the $T$ value corresponding to this point is very similar for 3x3x3 and 4x4x4 cubes and neural networks of different sizes and depths. Thus, the experiments above reveal a rather unexpected effect - performance stagnation with respect to the train size.

Additional tests were performed to check if the proposed neural network faces grokking when being trained on huge amounts of data. For this purpose, we spent more than 7 days fully utilizing two NVIDIA H100 GPUs to train two neural networks on 524B examples of 4x4x4 Rubik's cube states. In both cases, the loss was continuously decreasing from $\approx 26.4$ on 8B examples down to $\approx 23.8$ at the end of training. There was no sign of delayed generalization. At the same time, the loss was notably decreasing, which led us to further analysis of the stagnation effect.

Then, we used both of these networks to solve the complete set of Santa Challenge scrambles for the 4x4x4 cube. For fair comparison, we also used snapshots of both networks made after training on 8B examples to solve the same scrambles.

First, we faced the fact that, using half-precision during inference, the solvers based on networks trained on 524B examples were unable to solve any of the scrambles. On the contrary, the versions trained on 8B examples were fully compatible with half-precision inference. Thus, we faced the first drawback of using huge trainsets — the longer inference time.

The comparison of the solutions found using single-precision during inference shows that networks trained on 524B examples achieved average solution rates of 48.56 and 49.46, finding solutions for 43/43 and 41/43 scrambles, respectively. As seen from Figure 3, these results are comparable with the ones achieved by the solvers with neural networks trained on 8B of examples. The solvers equipped with snapshots of the tested networks, stored after training on 8B examples, achieved average solution rates of 49.3 and 49.74, finding solutions for 43/43 and 42/43 scrambles, respectively. From one point of view, the results provided by networks trained on 524B examples are generally better (even though one of the networks "forgot" how to solve one of the scrambles). From another point of view, during the computation time spent on such long training, up to 63 networks can be trained on 8B examples. At the same time, as it will be shown further (Figure 3), even ten solvers equipped with networks trained on 8B examples provide an average solution length far below 47. The detailed solution analysis for the experiment is provided in Appendix B.7.

The example described above illustrates our interpretation of the observed stagnation effect. It does not mean that further training will not provide a better average solution rate. However, at some point, the computational effort for further training exceeds that required to achieve the same or better results using multiple agents, which are based on networks trained on fewer examples. Moreover, multiple agents can be easily accelerated using distributed computing, whereas boosting the training and inference of a single-agent approach generally requires more advanced GPU hardware.

**MLP layers and sizes.** As expected, larger and deeper networks trained on train sets of the same size generally provide shorter solutions than smaller models. What is less expected is that the higher number of layers (higher $N_1$, $N_2$, and $N_r$) is more significant than a larger number of parameters $P$. More surprising is that even small models with 1M of parameters can reach the average solution length comparable to DeepCubeA and EfficientCube using neural networks with $\approx 14.7M$ parameters.

Based on these observations for further consideration, we used a deep neural network having the same number of layers (ten) as [3] and [46], but with the smaller model size of 4M parameters.

**Beam width.** It is the most important parameter. We performed multiple tests on a single agent equipped with this model, changing $W$ from $2^{12}$ to $2^{24}$. The results of these tests are presented in Figure 2b (the exact model parameters and details of the tests are provided in Appendix B.6). From Figure 2b, it is clear that increasing $W$ effectively reduces the average solution length. Moreover, the solution length decreases approximately linearly with the logarithm of the beam width $W$. On the 3x3x3 cube, increasing $W$ up to $2^{24}$ allows us to get close to the optimal solution, while for 4x4x4, the same beam width results in a better average solution length than the best ones submitted to the 2023 Santa Challenge.

**Agents number.** In the third part of the experimental studies, we investigated the influence of the number of agents $A$ on the solver's efficiency. These experiments were performed on 3x3x3, 4x4x4, and 5x5x5 Rubik's Cube. We used 10-layer ResMLP models with 4M parameters trained on 8B states in all the cases. The beam width was chosen $W = 2^{24}$ so each agent could fit into the memory of a single GPU regardless of the solved cube size. The details of the performed experiments are available in Appendix B.6, while their results are provided in Figure 3. For ease of analysis, Figures 3a,3b,3c demonstrate lengths only for those agents whose solutions at least once were used as the solver's output. The shaded area in Figure 3, representing the standard error of the mean (the standard deviation divided by the square root of the number of samples), calculated as the standard deviation divided by the square root of the number of scrambles used to compute each average, which allows comparison of solver performance.

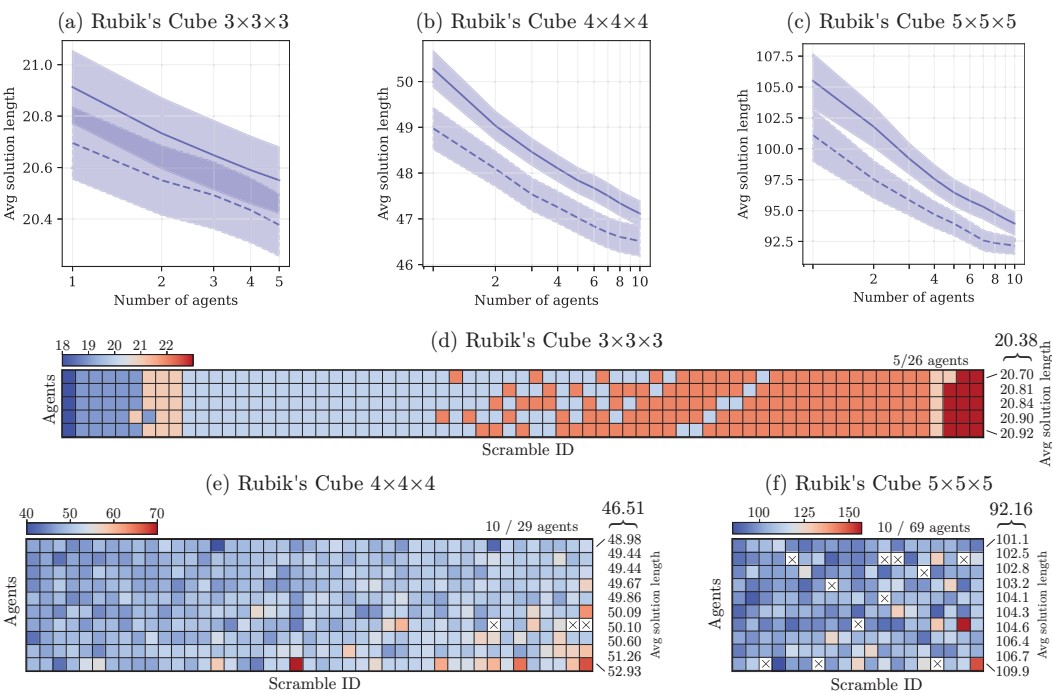

Figure 3: Average solution length of the proposed multi-agent approach depending on the number of agents composing its output for (a) 3x3x3, (b) 4x4x4, and (c) 5x5x5 Rubik's cubes. Solid line – random set of the agents, dashed line – best set. Shaded area indicates ±1 standard error around the mean solution length. Distribution of solution lengths for (d) 3x3x3, (e) 4x4x4, and (f) 5x5x5 Rubik's cubes for the best ensemble.

Figure 3 clearly shows that the average solution rate of a multi-agent is always higher than the one achieved by the best single agent (up to 8 moves for the 5x5x5 cube). Solid lines on Figures 3a,3b,3c show how the size of the ensemble influences the average solution length for the random set of the agents. As seen in all three cases of 3x3x3, 4x4x4, and 5x5x5 Rubik's cubes, the larger number of agents robustly provided more optimal pathfinding. The dashed line demonstrates the same dependency but for the set of agents jointly providing the best overall solution. Figures 3c,3d,3e

demonstrate in color code how each agent from this set participates in the final solution for every scramble from the dataset. The scrambles which were not solved in less than 200 moves are marked with crosses.

As seen from Figures 3c,3d,3e, the worst agents in the ensemble not only provide much longer results than the final solution but also include multiple scrambles that were unsolved. In the case of using the single-model approach, these agents would be considered unsatisfactorily trained. Nevertheless, they are included in the best ensemble because they provided the shortest solution on one or two scrambles. Moreover, our approach achieved an efficiency that was previously unreachable for other ML solutions, largely due to the specialized agents.

Even though the results presented in Figure 3 on 3x3x3, 4x4x4, and 5x5x5 cubes can be achieved using 5, 10, and 10 agents, respectively, the probability of training all these agents in a row is very low. For example, to beat all the 5x5x5 scrambles from the 2023 Santa Challenge dataset, we trained 69 different agents, while further analysis showed that only 10 of them composed all the output results. At the same time, the first agent trained to solve 4x4x4 cubes beat all the respective scrambles from the mentioned dataset but did not even get in the final ensemble because multiple other agents jointly surpassed it. Thus, achieving a high level of optimality requires many agents, as seen from the logarithmic nature of the plots demonstrated in Figure 3. Nevertheless, due to the high scalability of the proposed approach and the ability to run on distributed hardware using dozens of independent agents, it is not an issue using modern computational hardware.

## 4 Results Summary and Comparison with Prior Art

### 4.1 Comparison with the other solutions for Rubik's cube solving

Table 1[3] summarizes the main results achieved by the proposed solver, highlighting its superiority over the prior state of the art. Notably, it surpasses the 2023 Kaggle Santa Challenge results, where over a thousand teams competed in virtual puzzle solutions, representing the best available methods and results. It should be mentioned that we were limited in computation resources during our research. Thus, our results can be improved even more by using more advanced hardware, which will allow for an increase in beam width and the number of agents.

Table 1 contains the following abbreviations: HTM – half-turn metric, QTM – quarter-turn metric. The 2023 Kaggle Santa Challenge dataset uses modified QTM with unfixed corners and centers of the cube, which is marked UQTM. Optimal PDB+ solver is a solver used to determine the God's number for the 3x3x3 Rubik's Cube[4]. The results presented in [49] were evaluated using a single cube scrambled with 100 random moves. For these results, column "Avg. solution len" contains the minimal length achieved by the most suitable genetic algorithm configuration. The optimality was not evaluated for these results and the results from the 2023 Kaggle Santa Challenge dataset. The values provided in parentheses in Table 1 indicate the standard error of the mean. From the perspective of sampling random states of a given group, this allows statistically significant comparisons between the performance of different solvers.

A single agent with single-layer MLP can solve all the DeepCubeA dataset with 90.4% of optimality, significantly enhancing results of the most advanced state-of-the-art ML solutions: DeepCubeA and EfficientCube. 26 agents equipped with 10-layer ResMLP models managed to solve all 1000 scrambles from DeepCubeA dataset with 97.6% optimality, which is the best result ever achieved by any ML solution (significantly surpassing 60.3%, 69.8% results from DeepCubeA and EfficientCube). A single-agent solution implemented using our approach and 10-layer ResMLP managed to beat each best result corresponding to 3x3x3 and 4x4x4 Rubik's cubes submitted on the 2023 Kaggle Santa Challenge (averages: 48.98 vs 53.49). At the same time, 29 agents managed to solve all the 4x4x4 cube's scrambles from the 2023 Kaggle Santa Challenge dataset with an average solution length of 46.51 - which is below 48 (a conjectured 4x4x4 Rubik's cube diameter [51]). Finally, an ensemble of 69 agents beat each best solutions for the 5x5x5 Rubik's cube submitted to the 2023 Kaggle Santa Challenge, shortening the average solution rate among all the datasets on more than 4.4 units in QTM metrics (ours: 92.16, Santa: 96.58). It is worth emphasizing that the solutions that were obtained outperformed the Santa results on average and in every single case.

---

[3]All the solvers presented in the Table 1 managed to solve all the scrambles from the listed datasets.

[4]`https://github.com/rokicki/cube20src`

Table 1: The most notable results achieved by the proposed solution and comparison with competitors.

| No. | Solver | Metric, Size, Dataset | A | W | P | T | Average solution length | Optimality |
|---|---|---|---|---|---|---|---|---|
| | | | | | | | | |
| **2x2x2 Rubik's cube** | | | | | | | | |
| 1 | Genetic [49] | HTM, 1, [49] | n/a | n/a | n/a | n/a | 30 | n/a |
| 2 | Breadth First Search | QTM, 100, Ours | n/a | n/a | n/a | n/a | 10.7(1) | 100% |
| **3** | Ours, 1-layer MLP | QTM, 100, Ours | 1 | $2^{18}$ | 0.15M | 8B | 10.7(1) | 100% |
| **4** | Ours, 10-layer ResMLP | QTM, 100, Ours | 1 | $2^{18}$ | 0.92M | 8B | 10.7(1) | 100% |
| **3x3x3 Rubik's cube** | | | | | | | | |
| 5 | Genetic [49] | HTM, 1, [49] | n/a | n/a | n/a | n/a | 238 | n/a |
| 6 | Optimal PDB+ solver [3] | QTM, 1000, [3] | n/a | n/a | n/a | n/a | 20.64(3) | 100% |
| 7 | DeepCubeA [3] | QTM, 1000, [3] | 1 | n/a | 14.7M | 10B | 21.50(3) | 60.3% |
| 8 | EfficientCube [46] | QTM, 1000, [3] | 1 | $2^{18}$ | 14.7M | 52B | 21.26(3) | 69.6% |
| 9 | EfficientCube [46] (reproduced) | QTM, 1000, [3] | 1 | $2^{18}$ | 14.7M | 52B | 21.26(3) | 69.8% |
| **10** | **Ours**, 10-layer ResMLP | QTM, 1000, [3] | 1 | $2^{18}$ | 4M | 8B | **21.14(3)** | **75.4**% |
| **11** | **Ours**, 1-layer MLP | QTM, 1000, [3] | 1 | $2^{24}$ | 0.34M | 8B | **20.83(3)** | **90.4**% |
| **12** | **Ours**, 10-layer ResMLP | QTM, 1000, [3] | 1 | $2^{24}$ | 4M | 8B | **20.69(3)** | **97.3**% |
| **13** | **Ours**, multi-agent 10-layer ResMLP | QTM, 1000, [3] | 26 | $2^{24}$ | 4M | 8B | **20.67(3)** | **98.4**% |
| 14 | Santa Challenge [50] | UQTM, 82, [50] | n/a | n/a | n/a | n/a | 21.8(1) | n/a |
| **15** | **Ours**, 10-layer ResMLP | UQTM, 82, [50] | 1 | $2^{24}$ | 4M | 8B | **19.5(1)** | n/a |
| **4x4x4 Rubik's cube** | | | | | | | | |
| 16 | Genetic [49] | HTM, 1, [49] | n/a | n/a | n/a | n/a | 737 | n/a |
| 17 | Santa Challenge [50] | UQTM, 43, [50] | n/a | n/a | n/a | n/a | 53.5(2) | n/a |
| **18** | **Ours**, 10-layer ResMLP | UQTM, 43, [50] | 1 | $2^{24}$ | 4M | 8B | **49.0(4)** | n/a |
| **19** | **Ours**, multi-agent 10-layer ResMLP | UQTM, 43, [50] | 29 | $2^{24}$ | 4M | 8B | **46.5(3)** | n/a |
| **5x5x5 Rubik's cube** | | | | | | | | |
| 20 | Genetic [49] | HTM, 1, [49] | n/a | n/a | n/a | n/a | 1761 | n/a |
| 21 | Santa Challenge [50] | UQTM, 19, [50] | n/a | n/a | n/a | n/a | 96.6(8) | n/a |
| **22** | **Ours**, multi-agent 10-layer ResMLP | UQTM, 19, [50] | 69 | $2^{24}$ | 4M | 8B | **92.2(7)** | n/a |

Analyzing comparison results in Table 1 it should be noticed, that $A$ agent equipped with model of $P$ size are equivalent to single agent with model having $A \cdot P$ parameters (e.g., 26 agent with 4M models are computationally comparable to an agent with 104M model). At the same time, from hardware point proposed multi-agent solution is much easier to scale using distributed computing than approaches based on large single model.

## 4.2 Comparison with the previous art in terms of computation time

The efficiency of our approach is driven not only by the large number of agents but also by the efficiency and simplicity of each single node. It is difficult to fairly compare computation time between different solutions, because experimental studies published in research papers were performed using different hardware. Moreover, ins such direct comparison disadvantages of algorithm can be hidden by performance boost driven by more modern GPU family. Thus, we performed an additional test and compared it with EfficentCube, which is claimed to be more efficient than DeepCubeA [46] in terms of average computation time while running on the same hardware. The training procedure for EfficentCube took 25 hours 50 minutes 45 seconds, while the model for our solution was trained in 3 hours 2 minutes 36 seconds. In both cases training was performed using 32-bit floats. Then, both solutions were used to solve all the scrambles from the DeepCubeA dataset (see results No.9 and 10 in Table 1). For both solutions we used the same beam width of $2^{18}$, which provided very close average solution length. The similarity in solution length makes this comparison fair even both algorithms rely on different models and beam search implementations. Finally, EfficientCube required 237.9 s on average to solve a single scramble, while our solution required 9.16 s, which is $\approx 25.6$ times faster.

It is worth mentioning that comparison is fair not because we used the same beam width, but because in both cases solvers were running on the same hardware and provided very close average solution length. Multiple factors cause the demonstrated performance boost. First, we used a model containing 3.675 times fewer parameters and only one output (instead of the 12 outputs used in EfficientCube). This model generally requires less computational resources for both training and inference. Moreover, the training duration was shortened by 6.5 times less trainset size. Second, we used 16-bit half-precision floating point variables instead of 32-bit wide during inference. EfficientCube was configured with the same option, but due to inefficient software implementation it gives only 10% performance gain, while in our case the computation speed was approximately doubled. Third, our implementation is much more optimized for running on GPU, which provides the rest of the performance boost. Finally, our approach is much more scalable because agents are independent and can be executed on different processors and GPUs.

## 4.3 Analysis of efficiency for solving other puzzles

To demonstrate the generality of our approach beyond Rubik's cube groups, we conducted experiments across a diverse range of permutation groups with sizes up to $10^{145}$ elements. The details of this experiment and its results are presented in the Section B.8 and Table 4. In short, we fixed the neural network architecture parameters ($N_1 = 1024$, $N_2 = 256$, $N_r = 1$) and trained models for 128 epochs on various groups, using beam width $W = 2^{20}$ for all experiments. This experiment design reveals both the strengths and limitations of our approach—while some extremely large groups remain unsolvable with these fixed parameters, the method successfully solves most tested groups with remarkable efficiency—achieving short training times (typically under 2 minutes using), fast inference (under 1 minute for most groups), and producing solution paths of practical length [5].

For groups not included in the Santa Challenge, we tested performance on 10 states obtained through 10,000 random moves from the solved state. Our approach successfully solved 100% of test cases for 28 out of 39 tested groups, including challenging domains like Pancake sorting [52] (known as NP-hard), LRX [53], and the 15-puzzle with periodic boundary conditions [54].

Our method consistently solved the 10 non-Rubik groups, and the average solution lengths were better than the top solutions in the Santa Challenge. This evaluation provides strong evidence that our approach generalizes effectively across various permutation groups, demonstrating robustness and computational efficiency regardless of group structure or size.

---

[5]Mentioned above times were achieved using NVIDIA H100 GPU.

### 4.4 Limitations

This paper reports on purely empirical research. Our results demonstrate that the proposed method is effective for multiple groups and outperforms all previously published machine learning solutions for solving Rubik's cubes up to 5x5x5. However, direct application of our results to other tasks requires additional theoretical analysis, which is beyond the scope of this paper.

To the best of our knowledge, there is no optimal solver for 4x4x4 and 5x5x5 Rubik's cubes. Furthermore, the diameter of the corresponding graphs has not yet been precisely defined. Thus, we were limited to the available reference datasets in order to provide results that are comparable to those of the state of the art. It should be noted that the Kaggle Santa 2023 Challenge Dataset includes only a small number of examples for large groups (e.g., it includes only 19 scrambles of the 5x5x5 Rubik's Cube). Considering this, broader generalization of the results requires additional research.

## 5 Conclusion

The paper proposes a machine learning-based approach to the pathfinding problem on large graphs. Experimental studies demonstrate that it is more efficient than state-of-the-art solutions in terms of average solution length, optimality, and computational performance.

The key parts of the approach are multi-agency, neural networks predicting diffusion distance and beam search. Deeper neural networks better approximate the graph of the large Rubik's cubes, though, for the 3x3x3 case, even a single-layer network provides excellent results. At the same time, the effect of enlarging the training set is limited: the trainset above 8196M examples for the tested models has no practical reason, which allowed us to avoid additional time spent during the training. Conversely, raising the beam width effectively lowers the solution length and increases optimality.

The complete set of the proposed solutions allowed the creation of the multi-agent pathfinder, which managed to beat all the ML-based competitors: an agent equipped with single-layer MLP solved all the DeepCubeA dataset with 90.4% of optimality significantly enhancing results of the most advanced state of the art solutions: DeepCubeA and EfficientCube. 26 agents equipped with 10-layer ResMLP models managed to solve all scrambles from DeepCubeA dataset with 97.6% optimality, which is the best result ever achieved by any ML solution. Single-agent solutions implemented using our approach and 10-layer ResMLP beat all the best results corresponding to 3x3x3 and 4x4x4 Rubik's cubes submitted on the 2023 Santa Challenge. At the same time, six agents managed to solve all the 4x4x4 cube's scrambles from the 2023 Santa Challenge dataset with an average solution length below 48 (a 4x4x4 Rubik's cube diameter predicted in [51]). Finally, a composition of 69 agents beat all the best solutions for the 5x5x5 Rubik's cube submitted to the 2023 Santa Challenge, shortening the average solution rate among all the datasets on more than 4.4 units in QTM metrics.

Finally, our experimental studies demonstrated that even a single agent trained using a relatively small number of examples can robustly solve a broad range of puzzles represented by Cayley graphs of size up to $10^{145}$, confirming the generality of the proposed method.

## 6 Acknowledgments

We express our gratitude to Graviton for providing servers for research in the field of AI.

We are deeply grateful to the many colleagues who expressed their interest, participated in fruitful discussions and have contributed to the CayleyPy project at various stages of its development, including: M. Douglas, M. Gromov, S. Nechaev, V. Rubtsov, J. Mitchell, M. Kontsevich, Y. Soibelman, A. Soibelman, S. Gukov, A. Hayat, T. Smirnova-Nagnibeda, D. Osin, V. Kleptsyn, G. Olshanskii, A. Ershler, J. Ellenberg, G. Williamson, A. Sutherland, Y. Fregier, P.A. Melies, I. Vlassopoulos, F. Khafizov, A. Zinovyev, H. Isambert, T. Ruzaikin, A. Fokin, M. Goloshchapov, S. Fironov, A. Lukyanenko, A. Abramov, A. Ogurtsov, A. Trepetsky, A. Dolgorukova, S. Lytkin, S. Ermilov, L. Grunvald, A. Eliseev, G. Annikov, M. Evseev, F. Petrov, N. Narynbaev, S. Nikolenko, S. Krymskii, R. Turtayev, S. Kovalev, N. Rokotyan, G. Verbyi, L. Shishina, A. Korolkova, D. Mamaeva, M. Urakov, A. Kuchin, V. Nelin, B. Bulatov, F. Faizullin, A. Aparnev, O. Nikitina, A. Titarenko, U. Kniaziuk, D. Naumov, A. Krasnyi, S. Botman, R. Vinogradov, D. Gorodkov, I. Gaiur, I. Kiselev, A. Rozanov, K. Yakovlev, V. Shitov, E. Durymanov, A. Kostin, R. Magdiev, M. Krinitskiy and P. Snopov.

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

# B Appendix: Methods

## B.1 Source code

The original source code is attached to the paper as Supplementary Material. The last version can be found on GitHub: `https://github.com/khoruzhii/cayleypy-cube`.

## B.2 Cayley graphs and Rubik's cubes

Moves of Rubik's cube can be described by permutations (e.g., Chapter 5 [55], or Kaggle notebook "Visualize allowed moves"[6]). Taking all the positions as nodes and connecting them by edges, which differ by single moves, one obtains a Cayley-type graph for Rubik's cube. Solving the puzzle is equivalent to finding a path on the graph between nodes representing the Rubik's cube's scramble initial and solved state.

## B.3 Random walks and train set generation

The training set is generated by scrambling (i.e., applying random moves) the selected solved state and creating a set of pairs $(v, k)$, where $k$ is a number of scrambles, and $v$ is a vector describing the node obtained after $k$ steps. In other words, we consider random walks on the graph. The main parameters are $K_{max}$ and $K$, where $K_{max}$ is a maximal number of scrambles (length of random walk trajectory), while $K \cdot K_{max}$ is a number of nodes to generate.

In the current research, we used so-called non-backtracking random walks [56], that forbid scrambling to the state of the previous step. A PyTorch-optimized implementation of train set generation can be found in *trainer.py* in the code attached to this paper.

Current research does not investigate the influence of $K_{max}$ on the solver's performance. We used $K_{max} = 26$ for solvers targeted on 3x3x3 cubes, $K_{max} = 45$ – for 4x4x4 cubes, and $K_{max} = 65$ for 5x5x5 cubes.

## B.4 Neural Network and Training procedure

In this study, we used ResMLP, a generalized form of multilayer perceptrons as described in [3, 46]. Details of the architecture can be found in Figure 1b. The PyTorch implementation of ResMLP is available in *model.py* in the code attached to this paper.

The training procedure was performed using the Adam optimizer with a fixed learning rate of 0.001 and mean squared error as the loss function. A new dataset of 1M examples was generated before each training epoch. All models were pre-trained and remained unchanged during puzzle-solving. Training was conducted using 32-bit floating point precision, while inference used 16-bit floating point numbers to enhance computational efficiency. The PyTorch implementation of the training procedure is available in *trainer.py* in the code attached to this paper.

## B.5 Beam-search

Beam search is a simple but effective search procedure used for various optimization tasks [57], [58], [59] as well as to improve outputs of the modern transformer-based language models [60],[61],[62]. It has been used in EfficientCube [46] and by many participants of the Kaggle Challenge [50]. We implemented a modified version of traditional beam search, which uses hash functions to remove duplicates, reducing the computation complexity of the pathfinder. Finally, in all the experiments, the scramble was considered unsolved if the path to the solved state was not found in 200 beam search steps. Additionally, the algorithm stops if the beam vector contains only already visited graph nodes. A PyTorch-optimized implementation of the beam search can be found in *searcher.py* in the code attached to this paper.

---

[6]`https://www.kaggle.com/code/marksix/visualize-allowed-moves`

## B.6 Experiments design

All the experiments were conducted using software attached to this paper. The experiments targeting analysis of trainset size's influence on the solver's performance included solving 20 scrambles of both 3x3x3 and 4x4x4 Rubik's cubes using different models as beam search heuristics. For this experiment, we prepared 20 models, whose parameters are demonstrated in the first 20 rows of Table 2. Each model was trained during 16384 epochs. The snapshots of the model parameters were saved after 16, 64, 256, 1024, 4096, and 16384 epochs. Then, each model snapshot was integrated as a heuristic into beam search with $W = 2^{18}$, which was used to solve the first 20 scrambles from the dataset. DeepCubeA dataset [3] was used for 3x3x3 Rubik's cube, and the 2023 Kaggle Santa Challenge [50] dataset was used for 4x4x4 puzzle. The results achieved by each solver configuration on the corresponding dataset were averaged and analyzed as experimental results. Scrambles unresolved due to low beam width were excluded from the demonstration in Figure 2 to simplify the overall plot analysis.

Table 2: The parameters of neural networks used in current research

| No. | Cube | Metric | Layers | $N_1$ | $N_2$ | $N_r$ | $P$ | Result No. |
|---|---|---|---|---|---|---|---|---|
| 1 | 3x3x3 | QTM | 1 | 3050 | 0 | 0 | 1M | – |
| 2 | 3x3x3 | QTM | 2 | 850 | 850 | 0 | 1M | – |
| 3 | 3x3x3 | QTM | 6 | 800 | 340 | 2 | 1M | – |
| 4 | 3x3x3 | QTM | 10 | 430 | 300 | 4 | 1M | – |
| 5 | 3x3x3 | QTM | 1 | 12196 | 0 | 0 | 4M | – |
| 6 | 3x3x3 | QTM | 2 | 1841 | 1841 | 0 | 4M | – |
| 7 | 3x3x3 | QTM | 6 | 2000 | 697 | 2 | 4M | – |
| 8 | 3x3x3 | QTM | 10 | 700 | 643 | 4 | 4M | 10, 12, 13 |
| 9 | 4x4x4 | UQTM | 2 | 750 | 750 | 0 | 1M | – |
| 10 | 4x4x4 | UQTM | 4 | 530 | 470 | 1 | 1M | – |
| 11 | 4x4x4 | UQTM | 6 | 720 | 300 | 2 | 1M | – |
| 12 | 4x4x4 | UQTM | 10 | 500 | 266 | 4 | 1M | – |
| 13 | 4x4x4 | UQTM | 2 | 1730 | 1730 | 0 | 4M | – |
| 14 | 4x4x4 | UQTM | 6 | 1180 | 1024 | 1 | 4M | – |
| 15 | 4x4x4 | UQTM | 6 | 2000 | 628 | 2 | 4M | – |
| 16 | 4x4x4 | UQTM | 10 | 1010 | 592 | 4 | 4M | 18, 19 |
| 17 | 4x4x4 | UQTM | 6 | 2000 | 1126 | 2 | 8M | – |
| 18 | 4x4x4 | UQTM | 10 | 1540 | 850 | 4 | 8M | – |
| 19 | 4x4x4 | UQTM | 6 | 5000 | 1369 | 2 | 16M | – |
| 20 | 4x4x4 | UQTM | 10 | 5000 | 1062 | 4 | 16M | – |
| 21 | 2x2x2 | QTM | 1 | 1024 | 0 | 0 | 0.15M | 3 |
| 22 | 2x2x2 | QTM | 10 | 430 | 300 | 4 | 0.92M | 4 |
| 23 | 3x3x3 | QTM | 1 | 1024 | 0 | 0 | 0.34M | 11 |
| 24 | 3x3x3 | UQTM | 10 | 700 | 643 | 4 | 4M | 15 |
| 25 | 5x5x5 | UQTM | 10 | 1008 | 560 | 4 | 4M | 22 |

The first experiment's results are demonstrated in Figure 2a. Single layer MLPs for 4x4x4 Rubik's cube are not presented in Table 2 as during preliminary research solvers equipped with this type of model did not manage to solve any scramble before reaching the 200 steps limit.

During the second experiment, we used only 10 layer models with size 4M (models No.8 and 16 from Table 2). The first experiment's results did not show a significant effect of increasing $T$ (train size) from 4B to 16B. Thus, the second experiment used finer granularity with 4B, 8B, and 16B train sizes to select the appropriate size more precisely. Each snapshot of the models trained with the mentioned above train sets was integrated into solvers with $W$ of $2^{12}$, $2^{14}$, $2^{16}$, $2^{18}$, $2^{20}$, $2^{22}$, and $2^{24}$. Then, these solvers were used to unscramble the first 20 puzzles from the same dataset used in the first experiment. The results achieved by each solver configuration on the corresponding dataset were averaged. Unsolved scrambles were excluded from consideration in this experiment.

Table 3: Average solution length depending from trainset size $T$ and beam width $W$ for tested with puzzle win probability above 0.5.

| $T$ | 4B | | 8B | | 16B | |
|---|---|---|---|---|---|---|
| $W$ | **Win prob.** | **Avg. sol.** | **Win prob.** | **Avg. sol.** | **Win prob.** | **Avg. sol.** |
| **3x3x3 Rubik's solver results with win probability above 0.5** | | | | | | |
| $2^{12}$ | 1.00 | 22.15 | 1.00 | 22.05 | 1.00 | 22.3 |
| $2^{14}$ | 1.00 | 21.55 | 1.00 | 21.55 | 1.00 | 21.6 |
| $2^{16}$ | 1.00 | 21.25 | 1.00 | 21.15 | 1.00 | 21.2 |
| $2^{18}$ | 1.00 | 21.15 | 1.00 | 20.95 | 1.00 | 21 |
| $2^{20}$ | 1.00 | 20.75 | 1.00 | 20.85 | 1.00 | 20.9 |
| $2^{22}$ | 1.00 | 20.65 | 1.00 | 20.65 | 1.00 | 20.7 |
| $2^{24}$ | 1.00 | 20.65 | 1.00 | 20.65 | 1.00 | 20.65 |
| **Average solution** | **21.16** | | **21.12** | | **21.19** | |
| **4x4x4 Rubik's solver results with win probability above 0.5** | | | | | | |
| $2^{16}$ | 0.80 | 61.88 | 0.55 | 60.18 | 0.85 | 64.12 |
| $2^{18}$ | 1.00 | 58.1 | 0.85 | 56.7 | 0.95 | 58.47 |
| $2^{20}$ | 1.00 | 54.7 | 1.00 | 55.3 | 1.00 | 54.7 |
| $2^{22}$ | 1.00 | 51.9 | 1.00 | 52.8 | 1.00 | 52.3 |
| $2^{24}$ | 1.00 | 50.4 | 1.00 | 50.6 | 1.00 | 49.6 |
| **Average solution** | **55.4** | | **55.11** | | **55.84** | |

The results of the second experiment are demonstrated in Figure 2b. A deeper analysis of the experimental results shows that if we consider only $W$ values that give a puzzle winning probability above 50%, the agent with the model trained on 8B states has a slightly better average solution length than the competitors (Table 3). Thus, for the rest of the experiments, we used the 8B train set as a compromise between solver performance and training time.

The third experiment analyzed the influence of the number of agents $A$ on the solver's efficiency. We used models No.8, 16, and 25 for this experiment from Table 2. We trained each of these models multiple times during 8192 epochs. Then, each model was integrated into a dedicated agent. Due to computation limitations, we run only two agents in parallel at the same time, assuming that with more available GPU instances, it will be possible to compute all of them simultaneously. Finally, the total number of pretrained agents for 3x3x3 was 26, 4x4x4 – 29, and 5x5x5 – 69. As in previous experiments, the agents aimed to solve 3x3x3 cubes were tested on the scrambles DeepCubeA dataset, while the rest were verified using the 2023 Kaggle Santa Challenge dataset. Due to the large size of the DeepCubeA dataset, in the third experiment, we used a subset of 69 3x3x3 scrambles, which were considered most difficult during preliminary research. The results of the experiment are shown in Figure 3.

Due to historical reasons, scramble order and corresponding IDs demonstrated on Figures 3 and 4 are different from the original ones in the 2023 Kaggle Santa Challenge dataset. This difference does not affect research results in any way, but may be confusing, e.g., in case one compares our best results with their own. Thus, for reproducibility reasons, we added *paper/solver-scrambles* and *paper/figure-scrambles* folders to the Supplementary Materials, which contain generators and scrambles used in each specific experiment and the corresponding README file.

The authors of [3], along with the well-known DeepCubeA dataset, were using DeepCubeA$_h$ set containing the scrambles that are furthest away from the goal state, assuming these scrambles are more challenging to solve. At the same time, original DeepCubeA solutions were robustly and optimally solving them. During the current research, we found another subset of the DeepCubeA dataset containing 16 scrambles, which were not solved optimally during the experimental studies. We believe that a significantly rising number of agents will lead to finding solutions to all of them. However, the first element of this subset (scramble No.17 from original DeepCubeA) was never optimally solved in any of our attempts, even during preliminary research and experiments not covered by this paper. We believe that analyzing the scrambles in this subset will help us understand

why they are so hard to solve compared to the rest of the DeepCubeA data. Finally, this understanding will lead to the development of new, more efficient ML methods.[7] Thus, we decided to publish these 16 scrambles as a self-contained dataset accompanied by this paper.

The experimental results listed in Table 1 were achieved by solving all the scrambles from the corresponding dataset defined in the third column of Table 1 using the proposed solver. The key solver parameters are listed in the fourth column. The last column of Table 2 demonstrates for which results from Table 1 each model was used.

The next experiment conducted in the current research was aimed to compare computational efficiency with the EfficientCube [46] (a state-of-the-art solution claimed by its author to have better efficiency than DeepCubeA). For the fairness of comparison, we used a dedicated server equipped with two Intel Xeon Gold 6442Y 24-core processors running at 2600 Mhz, 256 GB DDR5 RAM, 512 GB SSD, and two GPUs NVIDIA H100 80GB. The server was running Ubuntu Linux 24.04.2 LTS, CUDA 12.4. The latest version (March 10th, 2024) of the EfficientCube was downloaded from the official GitHub repository[8] and configured according to the author's instructions to reproduce the results from [46]. Our solution was installed on the same server and configured with the beam width providing similar solution length (see results No.9 and 10 in Table 1). First, we sequentially trained a model for each solution and measured the time required for these procedures. Then, we tested both solvers on all the scrambles from the DeepCubeA dataset. Both solutions were configured to use 32-bit single-precision floats for training and 16-bit half-precision floats for inference. We recorded the solving time for each scramble and then averaged it among the whole dataset. Finally, we compared training time and average solving time between EfficientCube and our solution. During this experiment, only one solution was running on the server at the same time.

### B.7 Analysis of the solutions provided by neural networks trained on 524B examples

The experiment aimed to analyze the benefits and drawbacks of using extremely large trainsets generated by random walks in solving Rubik's Cube. Two 10-layer neural networks were trained for this purpose. The parameters of these neural networks are provided in line 16 of Table 2. The training procedure was equal to the one described in Section B.4. The training was performed during 524288 epochs, resulting in 524B examples. The training was performed on a server equipped with two NVIDIA H100 GPUs. Each GPU was used to train only one network. The overall training procedure took approximately 7 days and 17.5 hours. Then each network was used to solve 43 4x4x4 scrambles from the 2023 Santa Challenge Dataset using a single agent with a beam width of $2^{24}$. Additionally, two solvers were equipped with snapshots of the trained networks stored after 8192 training epochs. The detailed comparison of the experimental results is provided in Figure 4. Figure 4 uses the following color code. Blue color – solution lengths of the best results, which were provided for the corresponding scramble by one of the neural networks during the experiment. The green color indicates that at least one of the neural networks trained with 524B examples produced shorter results than both versions trained with 8B examples. The red color indicates a scramble for which at least one of the neural networks trained with 8B examples produced shorter results than both versions trained with 524B examples. Yellow color – scrambles for which the shortest solutions provided by neural networks trained with 8B and 524B were the same.

Figure 4: The solution lengths found by the solvers equipped with neural networks trained using 8B examples and their versions extensively trained with 524B examples compared to the best solutions found using networks trained with 8B examples (in all other experiments performed during research).

---

[7]A possible explanation is that the number of optimal solution paths for such cubes is lower than average or equal to one, making these paths more difficult to find.

[8]https://github.com/kyo-takano/efficientcube

The first of two networks managed to solve all the scrambles, both trained using 8B and 524B samples. Sixteen scrambles were solved with the same length. In nine cases, a network snapshot captured after 8B samples provided shorter solutions, and in 18 cases, the network trained on 524B samples provided shorter solutions. As a result, the average solution length decreased from 49.3 to 48.56 as a result of very long training.

The results demonstrated by the solvers based on the second neural network are more controversial. The version trained using 8B samples managed to solve scrambles No. 10, and No. 29, but failed to solve the scramble No. 37. Conversely, the version trained using 524B samples solved the scramble No. 37, but failed to solve scrambles No. 10 and No. 29, providing a worse win probability rate. The solver, equipped with the network trained on 524B samples, provided a better average solution length, but only if unsolved scrambles are entirely omitted. Moreover, the difference between versions trained using 8B and 524B samples is much less than the one observed with the first network. Direct comparisons of different scrambles show that in 9 cases, network snapshots captured after 8B samples provided shorter solutions. In 16 cases, networks trained on 524B samples provided shorter solutions, and in the remaining 18 cases, the solution length was the same.

The comparison with the shortest solutions found during the rest of the research showed that networks trained with 524B samples did not outperform the best results demonstrated by the networks trained using 8B samples (Figure 4) for any of the scrambles. In 22 out of 43 cases, one of the solvers equipped with networks trained during the discussed experiment provided solutions with the same length as the shortest ones previously found for the corresponding scrambles. In 8 of these cases, the shortest solution was provided by both versions trained with 8B and 524B samples, and in 4 more cases, the 8B version provided the shortest solution, while the further trained neural network failed (Figure 4).

Thus, even long training with 524B samples slightly improved average solution length, but the advances achieved are incomparable to the additional computational resources required. Another experiment provided in this paper demonstrates that 5-10 agents trained using 8B samples (40-80B in total) achieve a smaller average solution length than any of the two networks we trained on 524B samples. Moreover, comparing these networks with their snapshots captured after 8192 training epochs demonstrated that they not only provide new, shorter solutions but also degrade for other scrambles and even fully "forget" how to solve some of them.

It was also revealed that extensive training on a large number of examples leads to more computationally extensive inference. The solvers equipped with networks trained with 524B samples were not able to solve any scrambles if half-precision inference was used. Thus, we were forced to compute the inference for these networks with single precision, which required more time on the same hardware. Contrarily, the versions trained with 8B samples were fully compatible with the "half precision" trick, which is used to boost the solver's performance.

### B.8 Method generalization

To demonstrate the generality of the proposed method beyond Rubik's cubes, we tested it on a diverse set of permutation groups from the 2023 Kaggle Santa Challenge (excluding cubes larger than 6x6x6), as well as additional groups such as Pancake sorting [52], LRX [53], and the 15-puzzle with periodic boundaries [54]. Each model was trained with fixed neural network parameters ($N_1 = 1024$, $N_2 = 256$, $N_r = 1$) for 16 and 128 epochs. The solver used beam width of $W = 2^{20}$. The results of this experiment are summarized in Table 4. The values provided in parentheses in Table 4 indicate the standard error of the mean. From the perspective of sampling random states of a given group, this allows statistically significant comparisons between the performance of different solvers.

The only varying parameter was $K_{max}$, selected based on known [63] or estimated [51] graph diameters. In general practice, $K_{max}$ can be chosen automatically by scanning several candidate values. We performed four independent training and inference runs for each group, confirming robustness for groups consistently solved in all runs (marked as 100% in Table 4).

Initial states for additional groups were generated by performing 10,000 random moves from the solved state, followed by an optional random move with 50% probability to ensure parity variability. The total test set included 296 puzzles from the Santa challenge groups and 200 puzzles from additional groups (10 per group).

The comparison of the results achieved using a model trained for 16 and 128 epochs (marked in Table 4 with their trainset size $T$ 16M and 128M, respectively) shows that in most of the cases, larger 128M trainset provides shorter solution length and as a consequence – faster scramble solving time. At the same time, it is surprising that models trained using only 16M examples robustly solve all the test cases for 28 of 39 investigated groups. Moreover, the Globe 6x8 puzzle model trained for 16 epochs was more successful than the one trained for 128.

We evaluated the method by comparing the average solution length to the best Kaggle Santa solutions. The proposed approach provided shorter average solutions for 12 out of 14 non-cube 2023 Kaggle Santa Challenge groups, indicating strong generalization. Failures occurred when the beam search stagnated, detected by repeating vertex sets across consecutive search steps. Such stagnation occurred for some extremely large groups, indicating that while the method is scalable, each group's practical solvability limit varies, as evident from Table 4. It is also seen that large Rubik's cubes are harder than many other puzzles represented with groups of comparable or larger size. A model trained with 16M examples did not solve any of the scrambles used during the experimental studies, while the model trained for 128 epochs solved only 22% of them. The previous experiments demonstrated that a larger model trained with 8192M examples can robustly solve 100% of scramble. At the same time, the found solution won't be the shortest one. Using up to 69 agents was required to beat all the best 2023 Kaggle Santa Challenge. Thus, the fact that in the last experiment, our method did not solve any scrambles of 6x6x6 Rubik's cube does not mean that it is not suitable for this task. It just may require a larger size of the model and beam width. For example, in one of our preliminary experiments, a single agent equipped with a neural network trained on 8 million parameters, 1 billion examples, and a beam width of 16 million managed to solve 2 out of 6 6x6x6 Rubik's Cube scrambles from the Santa Challenge dataset. The solution lengths were more than 200, but they were found using a smaller trainset, the same beam width, and only twice the size of the models used to achieve the result demonstrated in Table 1 for the 5x5x5 Rubik's cube.

The scripts for reproducing the results presented in Table 4, along with the exact values of the parameter $K_{\max}$ used for each group, are provided in the supplementary materials (*traintest-tab4-rnd.sh* and *traintest-tab4-santa.sh*).

Additionally, by suggestion of one of the reviewers on the rebuttal stage, we extended our experiments to include classical 15-puzzle instances (in addition to the periodic boundary version reported in Table 4 in the originally submitted version of the paper). We tested our method on standard 4x4 (15-puzzle) and 5x5 (24-puzzle) sliding puzzles, comparing against both classical PDB-based solvers and learning-based approaches.

For the 4x4 puzzle, our method achieves 100% optimality with an average solution length of 52.02 moves, matching the performance of PDB-based optimal solvers and slightly outperforming DeepCubeA (99% optimality). Our solving time of 3.5 seconds is faster than DeepCubeA's 10.3 seconds, though naturally slower than PDB's near-instantaneous 0.002 seconds. Crucially, our training time is only 40 seconds compared to DeepCubeA's 36 hours—a dramatic reduction in computational requirements.

For the more challenging 5×5 puzzle, our approach solves instances in 11 seconds with 88% optimality, compared to DeepCubeA's 19.3 seconds with 97% optimality. Interestingly, while PDB guarantees optimal solutions, it requires 4239 seconds (over an hour) to solve 5x5 puzzles—demonstrating that our learned heuristic provides a practical speed-optimality tradeoff for larger instances. EfficientCube was unable to solve the 5x5 puzzle in our tests. The original paper also does not report that this solution can solve the 24-puzzle.

These results demonstrate that our method offers a compelling balance: training times measured in seconds rather than days, competitive solution quality, and practical solving speeds that scale better than exact algorithms for larger puzzles. While domain-specific solvers like PDB excel in minor instances where precomputation is feasible, our approach offers a more general and scalable alternative that requires minimal domain knowledge.

Table 4: Generalization performance across different puzzles.

| Group | Size | Training, min | | Solving, min | | Avg. solition len. | | Solved, % | |
|---|---|---|---|---|---|---|---|---|---|
| | | 16M | 128M | 16M | 128M | 16M | 128M | 16M | 128M |
| Cube 2x2x2 | $4 \times 10^6$ | 0.1 | 0.5 | 0.01 | 0.02 | 10.5(3) | 10.5(3) | 100 | 100 |
| Cube 3x3x3 | $4 \times 10^{19}$ | 0.1 | 0.8 | 0.18 | 0.17 | 21.5(1) | 20.5(1) | 100 | 100 |
| Cube 4x4x4 | $7 \times 10^{45}$ | 0.2 | 1.5 | 1.44 | 1.24 | 77(2) | 65(1) | 90 | 94 |
| Cube 5x5x5 | $3 \times 10^{74}$ | 0.2 | 1.5 | – | 7.14 | – | 126(2) | 0 | 22 |
| Cube 6x6x6 | $2 \times 10^{116}$ | 0.4 | 2.8 | – | – | – | – | 0 | 0 |
| Wreath 6x6 | $3 \times 10^6$ | 0.1 | 0.4 | 0.00 | 0.00 | 8.5(4) | 8.5(4) | 100 | 100 |
| Wreath 7x7 | $2 \times 10^8$ | 0.1 | 0.4 | 0.00 | 0.00 | 9.2(3) | 9.2(3) | 100 | 100 |
| Wreath 12x12 | $1 \times 10^{21}$ | 0.1 | 0.5 | 0.01 | 0.01 | 18.3(7) | 18.3(7) | 100 | 100 |
| Wreath 21x21 | $4 \times 10^{47}$ | 0.1 | 0.5 | 0.05 | 0.05 | 35(2) | 35(2) | 100 | 100 |
| Wreath 33x33 | $6 \times 10^{88}$ | 0.1 | 0.6 | 0.10 | 0.11 | 61(1) | 61.5(5) | 100 | 100 |
| Globe 1x8 | $2 \times 10^{35}$ | 0.1 | 0.8 | 0.46 | 0.43 | 43(2) | 44(2) | 100 | 100 |
| Globe 1x16 | $1 \times 10^{89}$ | 0.1 | 1.1 | 2.09 | 1.98 | 83.3(7) | 81(3) | 100 | 100 |
| Globe 2x6 | $7 \times 10^{24}$ | 0.1 | 0.6 | 0.13 | 0.12 | 21(1) | 21(1) | 100 | 100 |
| Globe 3x4 | $4 \times 10^{26}$ | 0.1 | 0.7 | 0.20 | 0.18 | 29(2) | 27(1) | 100 | 100 |
| Globe 6x4 | $7 \times 10^{40}$ | 0.1 | 0.7 | 0.48 | 0.43 | 44.0(7) | 42(1) | 100 | 100 |
| Globe 6x8 | $2 \times 10^{107}$ | 0.2 | 1.8 | 6.58 | 7.02 | 183(16) | 173(15) | 100 | 75 |
| Globe 6x10 | $1 \times 10^{145}$ | 0.2 | 1.8 | – | 6.94 | – | 143 | 0 | 100 |
| Globe 3x33 | $1 \times 10^{448}$ | 0.8 | 6.6 | – | – | – | – | 0 | 0 |
| Globe 8x25 | $3 \times 10^{633}$ | 1.5 | 12 | – | – | – | – | 0 | 0 |
| Puzzle 8 | $2 \times 10^5$ | 0.1 | 0.5 | 0.00 | 0.00 | 12.8(5) | 12.8(5) | 100 | 100 |
| Puzzle 15 | $1 \times 10^{13}$ | 0.1 | 0.6 | 0.05 | 0.05 | 31.0(6) | 31.0(6) | 100 | 100 |
| Puzzle 24 | $8 \times 10^{24}$ | 0.1 | 0.9 | 0.17 | 0.17 | 64.6(8) | 64.2(8) | 100 | 100 |
| Puzzle 35 | $2 \times 10^{41}$ | 0.2 | 1.2 | 0.56 | 0.45 | 115(2) | 113(2) | 70 | 100 |
| Puzzle 48 | $3 \times 10^{62}$ | 0.2 | 1.8 | 1.79 | 2.07 | 181 | 181 | 10 | 10 |
| Puzzle 63 | $9 \times 10^{88}$ | 0.3 | 2.5 | – | – | – | – | 0 | 0 |
| LRX 10 | $4 \times 10^6$ | 0.1 | 0.5 | 0.01 | 0.01 | 28(1) | 28(1) | 100 | 100 |
| LRX 15 | $1 \times 10^{12}$ | 0.1 | 0.6 | 0.07 | 0.07 | 66(3) | 66(3) | 100 | 100 |
| LRX 20 | $2 \times 10^{18}$ | 0.1 | 0.8 | 0.18 | 0.19 | 135(3) | 135(3) | 100 | 100 |
| LRX 25 | $2 \times 10^{25}$ | 0.1 | 1.0 | 0.42 | 0.42 | 216(7) | 217(7) | 100 | 100 |
| LRX 30 | $3 \times 10^{32}$ | 0.1 | 1.1 | 0.52 | 0.52 | 386(16) | 329(15) | 50 | 50 |
| Pancake 10 | $4 \times 10^6$ | 0.1 | 0.5 | 0.01 | 0.01 | 8.6(3) | 8.6(3) | 100 | 100 |
| Pancake 15 | $1 \times 10^{12}$ | 0.1 | 0.5 | 0.06 | 0.06 | 13.9(3) | 13.9(3) | 100 | 100 |
| Pancake 20 | $2 \times 10^{18}$ | 0.1 | 0.6 | 0.14 | 0.14 | 18.8(3) | 18.8(3) | 100 | 100 |
| Pancake 25 | $2 \times 10^{25}$ | 0.1 | 0.7 | 0.38 | 0.38 | 24.0(3) | 24.0(3) | 100 | 100 |
| Pancake 30 | $3 \times 10^{32}$ | 0.1 | 0.8 | 0.58 | 0.58 | 28.4(8) | 28.2(7) | 100 | 100 |
| Pancake 35 | $1 \times 10^{40}$ | 0.1 | 1.0 | 1.26 | 1.24 | 34.5(3) | 34.2(2) | 100 | 100 |
| Pancake 40 | $8 \times 10^{47}$ | 0.1 | 1.1 | 1.22 | 1.20 | 39.4(4) | 38.9(3) | 100 | 100 |
| Pancake 45 | $3 \times 10^{56}$ | 0.2 | 1.2 | 3.25 | 2.99 | 48(1) | 44.1(4) | 100 | 100 |
| Pancake 55 | $3 \times 10^{73}$ | 0.2 | 1.4 | 6.70 | 3.49 | 57.2(8) | 50.0(3) | 50 | 100 |

## B.9   Computational resources

All of the experiments reported in the paper were performed on a dedicated server manufactured by Graviton. This server is equipped with two Intel Xeon Gold 6442Y 24-core processors running at 2600 Mhz, 256 GB DDR5 RAM, 512 GB SSD, and two GPUs NVIDIA H100 80GB. The server was running Ubuntu Linux 24.04.2 LTS, CUDA 12.4. None of the other applications except the ones used for this research were executed during experimental studies. The server was simultaneously running two agents, each using a dedicated GPU. In cases when more agents were needed, they were executed sequentially by pairs. The total computation time used to perform the research, including all the published experiments, took approximately two months of continuous computing with 100% GPU usage. During our preliminary research, we also used several virtual machines provided by Kaggle[9].

---

[9]https://www.kaggle.com/

