# OpenReview forum: "A machine learning approach that beats Rubik's cubes"
_NeurIPS.cc/2025/Conference — NeurIPS 2025 spotlight_

### Official Review · Reviewer_Zg9J · 2025-07-02

**Clarity:** 2
**Significance:** 2
**Originality:** 3
**Rating:** 4
**Confidence:** 4

**Summary:**

This paper introduces a machine learning approach for solving large Rubik’s cubes by combining diffusion-distance learning via ResMLP networks with beam search. The method uses a multi-agent setup, where each agent is trained with a different randomized dataset generated through random walks on Cayley graphs. The proposed solver significantly outperforms prior machine learning methods on 3x3x3 cubes and is the first to generalize effectively to 4x4x4 and 5x5x5 puzzles, surpassing results from the 2023 Kaggle Santa Challenge. The method is computationally efficient and demonstrates robustness across a broad range of permutation group puzzles.

**Questions:**

1. Could you clarify why you do not compare against heuristic search baselines, such as IDA*, A*, or PDB-guided search, which are more appropriate comparisons for a beam search-based method?
2. Can your random-walk-based diffusion distance training approach be used to enhance other search heuristics e.g., A*, IDA*, or greedy best-first search, rather than just beam search? If not, what specific aspects of beam search make it uniquely compatible?
3. You include results on several classical combinatorial problems (e.g., 15-puzzle, 24-puzzle,...), could you clarify how your method compares, in terms of accuracy, solution quality (e.g., move count), and runtime, with classical solvers such as PDB-based IDA*, or learning-based solvers like DeepCubeA, on these domains?

**Ethical Concerns:**

["NO or VERY MINOR ethics concerns only"]

**Final Justification:**

As discussed, I’m most concerned about the importance of solving the large Rubik’s over the regular 3×3×3, so I’ll keep my score.

**Quality:**

3

**Strengths And Weaknesses:**

Strengths:
1. Well-structured combination of random walk-based diffusion distance learning and beam search.
2. Efficient multi-agent strategy, reducing training time and enabling parallelization.
3. Good performance across sizes.
4. The method is significantly faster.

Weaknesses:
1. No Comparison with Classical Heuristic Search (e.g., IDA*), while the method uses beam search as a core component, the evaluation omits comparisons with other heuristic search algorithms like IDA*, A*, or PDB-guided search. For a fair assessment, the proposed beam search + learned heuristic should be compared with classical search baselines that also use heuristics, not only RL agents.

2. The proposed approach relies on search to correct for errors in the learned distance function, whereas DeepCubeA and similar models are pure RL policies that do not search during inference. Comparing these directly can be misleading, search-based solvers should be evaluated against other search-based solvers, not pure policy networks.

---

> ### Author Rebuttal · Authors · 2025-07-29
>
> 1.1. No Comparison with Classical Heuristic Search (e.g., IDA*), while the method uses beam search as a core component, the evaluation omits comparisons with other heuristic search algorithms like IDA*, A*, or PDB-guided search For a fair assessment, the proposed beam search + learned heuristic should be compared with classical search baselines that also use heuristics, not only RL agents.
>
> 1.2. Could you clarify why you do not compare against heuristic search baselines, such as IDA*, A*, or PDB-guided search, which are more appropriate comparisons for a beam search-based method?
>
> 1.3. No Comparison with Classical Heuristic Search (e.g., IDA*), while the method uses beam search as a core component, the evaluation omits comparisons with other heuristic search algorithms like IDA*, A*, or PDB-guided search. For a fair assessment, the proposed beam search + learned heuristic should be compared with classical search baselines that also use heuristics, not only RL agents.
>
> **Reply:** We thank the reviewer for pointing out these concerns. We compared our result for 2x2x2 and 3x3x3 Rubik's Cubes with Breadth-first search and PDB+ search-based solver, respectively. Our solution demonstrated the best efficiency compared to other methods based on neural network heuristics. The comparison with PDB+ or other search-based solvers on larger puzzles was limited by computational resources. For example, following the suggestion of another reviewer, we added a new experiment directly comparing our solver with DeepCubeA and PDB-based solver on 15- and 24-puzzles. Our method solved the 15-puzzle in 3.5 seconds, DeepCubeA in 10.3 seconds, and the PDB-based solver in 0.002 seconds. At the same time, for the 24-puzzle, our approach solves instances in 11 seconds, DeepCubeA in 19.3 seconds, and PDB requires 4239 seconds for the same task. As seen from the example above, the PDB-solver's computational resource usage grows dramatically with the size of the puzzle. To the best of our knowledge, no PDB-based solutions for 4x4x4 or larger cubes were reported yet.  Thus, to fit into the limitation on available computation resources, we used the Santa Challenge dataset as a reference dataset for the cubes larger than 3x3x3.
>
> 2. The proposed approach relies on search to correct for errors in the learned distance function, whereas DeepCubeA and similar models are pure RL policies that do not search during inference. Comparing these directly can be misleading, search-based solvers should be evaluated against other search-based solvers, not pure policy networks.
>
> **Reply:** DeepCubeA and similar models are indeed RL-based in their concepts. At the same time, "under the hood," they also use different search methods based on heuristics. E.g., DeepCubeA uses batch weighted A* search, which directly follows from their paper (Agostinelli, Forest, et al. "Solving the Rubik’s cube with deep reinforcement learning and search." Nature Machine Intelligence 1.8 (2019): 356-363., page 357, column 1, paragraph 4) and their source code. Similarly, EfficientCube relies on beam search as the backbone of the proposed algorithms. We compared our results with multiple methods, including genetic and search-based algorithms. E.g., we compared our result for 2x2x2 and 3x3x3 Rubik's Cubes with Breadth-first search and PDB+ search-based solver, respectively. Our solution demonstrated the best efficiency compared to other methods based on neural network heuristics. The comparison with PDB+ or other search-based solvers on larger puzzles was limited by our computational resources. To the best of our knowledge, no PDB solutions for 4x4x4 or larger cubes were reported yet. Thus, we used the Santa Challenge dataset as a reference dataset.
>
> 3. Can your random-walk-based diffusion distance training approach be used to enhance other search heuristics e.g., A*, IDA*, or greedy best-first search, rather than just beam search? If not, what specific aspects of beam search make it uniquely compatible?
>
> **Reply:** We thank the reviewer for highlighting this interesting question for discussion. We have developed a highly efficient GPU-based implementation of beam search, the source code of which is provided along with our paper. Thanks to its performance, we have managed to complete a large number of computation experiments using, in most cases, a single server with only two GPU accelerators. Since our research is purely empirical (which we directly state in our limitations, Section 4.4), it was crucial to provide comparability between all the experimental results. Therefore, we employed the same search method for all experiments. During the preliminary phase of our research, we performed several experiments on combining our approach with the implementation of the A* algorithm but did not achieve the desired performance. Then, when our method surpassed the results of DeepCubeA, which internally uses batch weighted A* search, we decided to focus our efforts on analyzing other factors affecting solution length using beam search for all the remaining experiments.
>
> 4. You include results on several classical combinatorial problems (e.g., 15-puzzle, 24-puzzle,...), could you clarify how your method compares, in terms of accuracy, solution quality (e.g., move count), and runtime, with classical solvers such as PDB-based IDA*, or learning-based solvers like DeepCubeA, on these domains?
>
> **Reply:** We thank the reviewer for raising this important question. Following your suggestion, we extended our experiments to include classical 15-puzzle instances (in addition to the periodic boundary version reported in Table 4). We tested our method on standard 4×4 (15-puzzle) and 5×5 (24-puzzle) sliding puzzles, comparing against both classical PDB-based solvers and learning-based approaches.
>
> For the 4×4 puzzle, our method achieves 100\% optimality with an average solution length of 52.02 moves, matching the performance of PDB-based optimal solvers and slightly outperforming DeepCubeA (99\% optimality). Our solving time of 3.5 seconds is faster than DeepCubeA's 10.3 seconds, though naturally slower than PDB's near-instantaneous 0.002 seconds. Crucially, our training time is only 40 seconds compared to DeepCubeA's 36 hours—a dramatic reduction in computational requirements.
>
> For the more challenging 5×5 puzzle, our approach solves instances in 11 seconds with 88\% optimality, compared to DeepCubeA's 19.3 seconds with 97\% optimality. Interestingly, while PDB guarantees optimal solutions, it requires 4239 seconds (over an hour) to solve 5×5 puzzles—demonstrating that our learned heuristic provides a practical speed-optimality tradeoff for larger instances. EfficientCube was unable to solve the 5×5 puzzle in our tests, the original paper also does not report the this solution is able to solve 24-puzzle.
>
> These results demonstrate that our method offers a compelling balance: training times measured in seconds rather than days, competitive solution quality, and practical solving speeds that scale better than exact algorithms for larger puzzles. While domain-specific solvers like PDB excel on smaller instances where precomputation is feasible, our approach provides a more general and scalable alternative that requires minimal domain knowledge.
>
> We will include these clarification in the Appendix of our camera ready version and incorporated classical puzzle results in Table 4 to provide a more comprehensive evaluation across different puzzle domains.

---

> > ### Comment · Reviewer_Zg9J · 2025-07-31
> > **Reply**
> >
> > Thank you for your response,  your paper title is:
> >
> > `A machine learning approach that beats large Rubik's cubes`
> >
> >
> > However, it turns out that on the 3 × 3 × 3 setting, the comparison is made against other NN-based methods, and it outperforms those — not the search-based methods like PDB+. Am I understanding that correctly?

---

> > > ### Author Response · Authors · 2025-08-01
> > >
> > > Thank you for pointing out this concern. Indeed, we are not competing with PDB+ or other search-based solvers on the 3x3x3 Rubik’s cube puzzle. PDB+ solver used as a reference is optimal. It is important to clarify that we intentionally did not claim 100% optimality so as to avoid any confusion suggesting that our approach is truly optimal. Currently, we provide 98.4% of optimality, which is a huge step from the previous state-of-the-art methods, based on artificial intelligence technologies (specifically, the ones based on neural networks and genetic algorithms). At the same time, it is worth noticing that we are considering 3x3x3 as a standard cube, while the main focus of our paper is to provide a unified method for solving larger cubes (4x4x4 and higher). To the best of our knowledge, there is no optimal search-based solver that were practically implemented for these types of Rubik’s cube. The main reason is exponentially growing computation resources, required to solve these cubes with the PDB search-based solvers. Also, as we know, PDB-based methods cannot be applied for solving Rubik’s cubes with even sides (e.g., 4x4x4). The main state-of-the-art competitors are the solutions provided by the participants of Kaggle machine learning challenge. However, this is not one method, but a set of different approaches, some of which were applied to solve only several specific scrambles. Still, we managed to surpass all of them. Thus, in our opinion, our title correctly reflects our main result: A **machine learning** approach that beats **large** Rubik's cubes.

---

### Official Review · Reviewer_yeSg · 2025-07-03

**Clarity:** 3
**Significance:** 4
**Originality:** 4
**Rating:** 5
**Confidence:** 4

**Summary:**

This paper proposes a novel ML-based approach for solving pathfinding problems on large Cayley graphs, with a focus on Rubik’s Cubes up to 5x5x5. It uses a ResMLP network trained to estimate “diffusion distance” from the solved state via random walks. This heuristic guides a beam search algorithm. A multi-agent system of independently trained models is used, with the best solution among them selected. The method achieves state-of-the-art results on 3x3x3–5x5x5 cubes and shows generality across other permutation puzzles.

**Questions:**

1. What is the root cause of this counterintuitive behavior, "performance stagnation" anomaly, where more training data leads to worse solver performance? Does this imply that for a certain scale, the "diffusion distance" estimation, based on random walks, becomes a less effective or even misleading heuristic for finding shortest paths, perhaps due to increasing noise or a fundamental disconnect from God's number in denser parts of the state space? Instead of random walks, it may be worth exploring alternative problem generators that more closely approximate optimal paths, as used in prior work [1, 2], to see if they provide better training signals and improve solver performance.


**References:**

[1] Büchner, C., Ferber, P., Seipp, J., & Helmert, M. (2022). *A Comparison of Abstraction Heuristics for Rubik's Cube*. ICAPS Workshop.

[2] Muppasani, B., Pallagani, V., Srivastava, B., & Agostinelli, F. (2024). *Comparing Rubik’s Cube Solvability in Domain-Independent Planners*.

**Ethical Concerns:**

["NO or VERY MINOR ethics concerns only"]

**Final Justification:**

I will stick to my score. The author's explanation of my concerns regarding the unexplained anomaly (performance stagnation, diffusion distance estimation) is convincing.

**Limitations:**

1. The authors explicitly state that their paper reports on "purely empirical research" . This means their findings are based on experimental observations rather than formal mathematical proofs or theoretical derivations.

**Paper Formatting Concerns:**

1. References should be started on a new page (10). Use the \newpage command in LaTeX.

**Quality:**

4

**Strengths And Weaknesses:**

**Strengths:**
1. The proposed solution shows state-of-the-art results and surpasses the aggregated best results from the Kaggle Santa 2023 challenge for 4x4x4 and 5x5x5 Rubik’s Cubes. For the 3x3x3 cube, it achieves an optimality rate exceeding 98%, matching task-specific solvers and significantly outperforming previous ML solutions like DeepCubeA (60.3%) and EfficientCube (69.6%).
2. The solution is approximately 25.6 times faster in solving 3x3x3 Rubik’s Cubes and requires up to 8.5 times less model training time than the most efficient state-of-the-art competitor like EfficientCube.
3. The use of multiple agents improves the average solution length. Each agent, trained on a different random walk-generated dataset, approximates the distance differently, and combining their results by selecting the shortest path found provides a more robust solution.

**Weakness/Suggestions:**
1. The paper reports a surprising result where models trained on a massive dataset (524B examples) failed to solve any 4x4x4 scrambles from the Kaggle Santa 2023 Challenge, while the same models trained on a much smaller dataset (8B examples) performed significantly better. This counter-intuitive performance stagnation is acknowledged but not explained, as the study remains largely empirical. The lack of theoretical analysis leaves open questions about whether overfitting to diffusion distance, noise accumulation, or other factors are responsible. Understanding why more data leads to worse performance is crucial, especially for scaling the method to larger problems.
2. The neural network is trained to predict "diffusion distance," an estimate based on random walks. While empirically effective, the theoretical connection between this "diffusion distance" and the true shortest path length (God's number) is not fully elaborated. The paper notes that beam search "compensates for any possible incorrectness in the neural network predictions", but a deeper understanding of why this specific heuristic guides optimal pathfinding so effectively, or why it fails at larger data scales, is not provided.
3. For large groups, such as the 5x5x5 Rubik's Cube, the Kaggle Santa 2023 Challenge Dataset only includes a small number of examples (e.g., 19 scrambles), which limits the broader generalization of the results.
4. The use of random walks to generate training data can introduce noisy or suboptimal learning signals, making the diffusion distance heuristic less reliable, especially at greater search depths. This may explain why the method requires a large number of agents to achieve high optimality. The paper itself notes that even the "worst agents" contribute uniquely, implying that diversity across independently trained models compensates for weaknesses in individual heuristics. However, with a more efficient problem generator—one that better aligns scramble depth with optimal solution length—fewer agents might be sufficient, potentially improving both training efficiency and convergence.

---

> ### Author Rebuttal · Authors · 2025-07-29
>
> 1. The paper reports a surprising result where models trained on a massive dataset (524B examples) failed to solve any 4x4x4 scrambles from the Kaggle Santa 2023 Challenge, while the same models trained on a much smaller dataset (8B examples) performed significantly better…
>
> **Reply:** We agree with the reviewer that the lack of theoretical proofs and analysis is a key limitation of our paper (we directly state this in Section 4.4 of our paper). The reported example, where models trained on a massive dataset fail to solve any of the scrambles, is explained by the fact that models trained with extremely large training sets start forgetting states around the goal, shifting their distance-to-goal predictions closer to the average distance among all states. Thus, the solver has no chance to find the target value based on such a heuristic. Corresponding clarifications will be added to the camera-ready version of the paper. At the same time, due to the empirical nature of our study, it cannot be guaranteed that this explanation covers all similar cases. It is worth noting that in the submitted paper, this example was used to illustrate the fact that a monotonically decreasing loss function during the training does not guarantee shorter solutions. Nevertheless, we recognize the importance of theoretical statements that accurately explain the observations reported in this paper, and we are currently working on developing them. At the same time, currently they are out of the scope of this research, which was aimed at significantly enhancing results achieved in prior art and demonstrating that ML-based solutions not relying on human knowledge in group theory can successfully solve a large-scale puzzle (e.g., 5x5x5 Rubik's cube) at a high level of optimality.
>
> 2. The neural network is trained to predict "diffusion distance," an estimate based on random walks. While empirically effective, the theoretical connection between this "diffusion distance" and the true shortest path length (God's number) is not fully elaborated…
>
> **Reply:** We agree with the reviewer that a deeper theoretical connection between "diffusion distance" and the true shortest path length (God's number) will help to understand how our solution can be improved or why it fails at larger data scales. The answer to the last reviewer’s comment contains additional clarification on this topic. Also submitting our paper to NeurIPS 2025, we performed several additional experiments on solving a 6x6x6 Rubik's Cube with our approach. A single agent equipped with a neural network trained on 8 million parameters, 1 billion examples, and a beam width of 16 million managed to solve 2 out of 6 scrambles from the Santa Challenge dataset. The solution lengths were more than 200, but they were found using the same beam-width, only twice the size of the models used to solve 5x5x5. Moreover, the amount of training data was even smaller than that used in the experimental studies reported in our paper. Figure 2 and Table 3 demonstrate that for the 4x4x4 cube, the results obtained using the 1B-16B trainset are comparable, which suggests that extremely large trainsets are not required to solve large puzzles. Interestingly, our new result on 6x6x6 does not contradict this assumption. As we have already addressed the previous comment, we are currently actively searching for a solid theoretical basis that will explain our observations and enable us to analytically predict the neural network parameters, training set size, and beam width required to solve large puzzles with an average solution length close to optimal. Meanwhile, this work is far from finished. Still, we believe that our observations, reported in our paper, will be of interest to many researchers working in the field of AI/ML applications in mathematics. We hope that this interest will convert into finding the discussed theoretical basis in a shorter time.
>
> **PS:** While answering this comment, we found a typo in Table 3. $T$ should be 4B, 8B, and 16B instead of 4M, 8M, and 16M. We will fix it in the camera-ready version of the paper.
>
> 3. For large groups, such as the 5x5x5 Rubik's Cube, the Kaggle Santa 2023 Challenge Dataset only includes a small number of examples (e.g., 19 scrambles), which limits the broader generalization of the results.
>
> **Reply:** We thank the reviewer for pointing out this concern. We specifically highlighted it in Section 4.4. For the smaller puzzles, such as the 3x3x3 Rubik's Cube, there are optimal algorithmic solvers (like PDB-based), which makes it easy to generate datasets containing optimal ground truth solution lengths of any size. Unfortunately, to the best of our knowledge, there are no optimal solvers available for 4x4x4 and larger cubes. As demonstrated in Table 4 (Appendix A.6), our solution can solve 5x5x5 cubes in a matter of minutes. However, the length of these solutions is far from the shortest one presented in Table 1. As the paper concludes, achieving shorter solutions generally requires increasing model depth, beam width, and the number of agents. The absence of optimal solvers significantly challenges the fair analysis of the proposed method's efficiency. Moreover, most of the published ML solutions were never implemented for the Rubik's Cube larger than 3x3x3. Thus, the 2023 Santa Challenge Dataset was the only one we found, which contained some sort of ground truth (the best solutions among over a thousand teams of ML developers competed to find the shortest puzzle solutions), which was possible to use for a fair assessment of our solution's efficiency.
>
> 4. The use of random walks to generate training data can introduce noisy or suboptimal learning signals, making the diffusion distance heuristic less reliable, especially at greater search depths. This may explain why the method requires a large number of agents to achieve high optimality. The paper itself notes that even the "worst agents" contribute uniquely, implying that diversity across independently trained models compensates for weaknesses in individual heuristics. However, with a more efficient problem generator—one that better aligns scramble depth with optimal solution length—fewer agents might be sufficient, potentially improving both training efficiency and convergence.
>
> **Reply:** We thank the reviewer for the valuable suggestion. Our team is currently investigating various approaches to trainset generation, aiming to make our random walks "smarter" and more efficient for the specific problem. We hope to present a dedicated paper on this topic shortly. At the same time, many possible random walk modifications that improve solver results rely on different human knowledge of the puzzle. At the same time, this paper aims to develop an ML solution that learns to solve specific puzzles based on demonstrated examples, without prior knowledge of the graph representing the puzzles or the laws describing their interconnections. Thus, in this research, we used the simplest version of random walks and demonstrated that, even with relatively small noisy trainsets, it is possible to "teach" our solution how to solve puzzles represented by very large graphs.
>
> 5. What is the root cause of this counterintuitive behavior, "performance stagnation" anomaly, where more training data leads to worse solver performance? Does this imply that for a certain scale, the "diffusion distance" estimation, based on random walks, becomes a less effective or even misleading heuristic for finding shortest paths, perhaps due to increasing noise or a fundamental disconnect from God's number in denser parts of the state space? Instead of random walks, it may be worth exploring alternative problem generators that more closely approximate optimal paths, as used in prior work [1, 2], to see if they provide better training signals and improve solver performance.
>
> **Reply:** We acknowledge that the root cause of this performance stagnation anomaly remains an open question that we are actively investigating. Regarding the effectiveness of diffusion distance as a heuristic, our preliminary investigations suggest that while the analytical diffusion distance can exhibit non-monotonic behavior with respect to the true distance, these non-monotonicities do not manifest as deep local minima that would fundamentally impair the solver's performance. It is also important to note that our approach does not simply converge to the raw diffusion distance; instead, it combines diffusion distance with a neural network model trained to approximate it. The interplay between these components, along with hyperparameters such as learning rate and regularization, can significantly influence the algorithm's behavior. It is a topic that warrants careful investigation in future work. We agree that denser parts of the state space pose the most significant challenge for our method, which may contribute to the performance stagnation at larger training scales. Regarding alternative problem generators, we concur that exploring more sophisticated random walk strategies could yield better training signals. We are actively developing approaches that could sample states more efficiently while maintaining good coverage of the state space. One promising direction involves biasing the random walks using easily computable heuristics (such as Manhattan or Hamming distance) to preferentially sample states that increase the heuristic value, controlled by a temperature parameter. However, for this initial study, we deliberately chose the simplest approach to isolate and demonstrate the effectiveness of the diffusion distance concept and multi-agent ensemble strategy in their pure form, allowing us to establish a clear baseline for future improvements.

---

> > ### Comment · Reviewer_yeSg · 2025-08-09
> > **Thank you**
> >
> > I thank the authors for their detailed rebuttal. Your hypothesis for the performance stagnation anomaly, *that models trained on massive sets "forget" states near the goal*, is a key insight. A follow-up thought on that: Have you considered if the network's parameter count is a contributing factor?  The network may be under-parameterized for a 524B-sample dataset, forcing it to learn a smoothed-out, average heuristic instead of capturing the sharp gradient needed near the solved state.  Adding this full context would strengthen the paper.

---

> > > ### Author Response · Authors · 2025-08-09
> > >
> > > That was one of our first thoughts when we deeply investigated this case, answering your initial comment. At the same time, results presented in Figure 2 demonstrate that models of different sizes in the range from 1M to 16M parameters face stagnation on similar trainset sizes. However, deeper and larger models generally stagnate, providing shorter average solution lengths.
> > >
> > > Currently, our main conclusion is that single agents trained using large trainsets are less effective in terms of achieving record-breaking short solutions than several agents trained on smaller trainsets. At the same time, following your comment, we are currently running several additional experiments, and if they provide any additional insights on the discussed phenomenon, we will add them to the camera-ready version of the paper.

---

### Official Review · Reviewer_D1fw · 2025-07-03

**Clarity:** 4
**Significance:** 3
**Originality:** 3
**Rating:** 5
**Confidence:** 5

**Summary:**

This paper shows that coupling multi-agent neural heuristics trained on random-walk data with beam search can solve the Rubik’s Cube and related combinatorial puzzles efficiently. The experimental evaluation is extensive and comprehensive, offering a valuable case study on how an ensemble of learned heuristics can be integrated with a classical graph-search procedure. The results markedly outperform recent baselines—most notably DeepCubeA and EfficientCube—in both solution quality and speed, underscoring the strength of the proposed approach.

**Questions:**

I believe I now have a solid grasp of the manuscript and therefore have no substantive questions. Nevertheless, I noticed several places that appear to contain typographical or labeling errors. Please verify the items below.

- Lines 217–218.
Original text: “The values provided in parentheses in Table 1 indicate the standard error of the mean.”
The figures in parentheses do not look like standard errors.

- Figure 1 caption.
Panels (b) and (c) appear to be swapped.

- Line 122. Figure 1c → Figure 1.

- Line 187. If it is seen … → As is seen …

- Line 205.
“Superiority → superiority.”

**Ethical Concerns:**

["NO or VERY MINOR ethics concerns only"]

**Final Justification:**

The authors have addressed my concerns. The first weakness I noted was minor, and their rebuttal is convincing. The second was a formatting issue that they have agreed to fix.

My other comments are limited to typos, which I trust will be corrected.

Therefore, I will maintain my original rating.

**Limitations:**

Section 4.4 provides an adequate explanation.

**Paper Formatting Concerns:**

There is no concern about the paper formatting.

**Quality:**

4

**Strengths And Weaknesses:**

Strengths
- Straightforward yet powerful approach.
The method is conceptually simple—train multiple lightweight MLP heuristics on random-walk data and fuse them via beam search—yet it proves highly effective in practice.

- Extensive, comprehensive experiments.
  The authors run a broad battery of tests across the standard 3×3×3 Rubik’s Cube and several harder combinatorial puzzles, making the paper a valuable case study for researchers exploring hybrid machine-learning + graph-search techniques.

- Impressive empirical performance.
  Compared with recent baselines such as DeepCubeA and EfficientCube, the proposed system uses a smaller network, requires less training time, searches faster, and still finds higher-quality solutions, albeit at the cost of running a multi-agent (parallel) beam search.

Weaknesses
- Limited novelty in ensemble idea.
It is not surprising that ensembling several independently trained random-walk models improves accuracy; this strategy is well-known in the literature. Nonetheless, the authors’ quantitative analysis of ensemble size vs solution quality offers useful insight, so the lack of conceptual novelty is a relatively minor drawback.

- Page 2 footnote is quite long; consider integrating its content into the main text to improve readability.

---

> ### Author Rebuttal · Authors · 2025-07-29
>
> 1. Limited novelty in ensemble idea. It is not surprising that ensembling several independently trained random-walk models improves accuracy; this strategy is well-known in the literature. Nonetheless, the authors’ quantitative analysis of ensemble size vs solution quality offers useful insight, so the lack of conceptual novelty is a relatively minor drawback.
>
> **Reply:** We agree with the reviewer that the ensemble concept is well-known. At the same time, in the discussed task, it was one of the key solutions that significantly improved the solver's results and surpassed all previous art ML solutions in terms of their length on Rubik's cube problems up to 5x5x5. Thus, we decided to stress the reader's attention on that fact. An interesting observation was also made that the ensembles that achieved the shortest results on the whole dataset included specialized agents, whose average solution length was not the best. At the same time, their combination resulted in solving each of the 5x5x5 scrambles in the competition dataset better than the best results achieved during the Kaggle Challenge. Finally, it is worth noting that the ensemble idea is often implemented as an ensemble of neural networks. We attempted this approach to predict the distance to the goal more accurately, but did not achieve any superior results. At the same time, an ensemble of solvers enabled us to outperform our competitors. Thus, we believe that the details of the ensemble idea implementation applied to our puzzle-solving problem are a valuable contribution that will help other researchers.
>
> 2. Page 2 footnote is quite long; consider integrating its content into the main text to improve readability.
>
> **Reply**: We thank the reviewer for the valuable suggestions. We will integrate the footnote into the main text in the camera-ready version of the paper.
>
> 3. Lines 217–218. Original text: “The values provided in parentheses in Table 1 indicate the standard error of the mean.” The figures in parentheses do not look like standard errors.
>     Figure 1 caption. Panels (b) and (c) appear to be swapped.
>     Line 122. Figure 1c → Figure 1.
>     Line 187. If it is seen … → As is seen
>     Line 205. “Superiority → superiority.”
>
> **Reply**:  We thank the reviewer for their assistance in helping us improve our paper. We will fix the listed typos and formatting issues in the camera-ready version of the paper.

---

> > ### Comment · Reviewer_D1fw · 2025-08-08
> >
> > Thank you for your comment.
> >
> > To be clear, the "weakness" I mentioned was intended more as a minor observation, raised because a reviewer is generally expected to identify at least one area for potential criticism. It is not something I consider a significant flaw in your work.
> > On the contrary, I think your paper shows very useful results. It is impressive that a simple idea can lead to such a good outcome, and I consider this to be a strong contribution.

---

> > > ### Author Response · Authors · 2025-08-08
> > >
> > > We thank you for your high assessment of our research.

---

### Decision · Program_Chairs · 2025-09-17

**Decision:**

Accept (spotlight)

**Comment:**

The paper presents a machine learning-based approach to the pathfinding problem by leveraging diffusion distance estimation and using beam search for pathfinding. Experiments show that the method's efficiency solves 4x4x4 and 5x5x5 Rubik's cubes with short solution lengths, outperforming existing solvers. On the 3x3x3 Rubik's cube, it achieves an optimality rate exceeding 98%, matching task-specific solvers and outperforming previous solutions like DeepCubeA (60.3%) and EfficientCube (69.6%).

Overall, the results are significant, and all reviewers recognized that this work provides strong empirical contributions. One issue raised was the lack of comparison with classical heuristic solvers such as IDA* and PDB. The authors acknowledged that such solvers are effective on smaller problems but become computationally infeasible for larger cubes. Still, it would have been better to provide some direct comparisons. I also agree with reviewer Zg9J that the use of "large Rubik's cubes" in the title is imprecise. While I understand that the authors explained "large" to refer to the complexity of the Cayley graphs, using the term in the title without additional context can be misleading. The authors should revise the title in the final version by either replacing or removing the word "large".

Nevertheless, the empirical results are really impressive. Hence, I am happy to recommend acceptance with a spotlight.